physiology

flatfish, teleost, aquaculture, gamete, artificial fertilization, assisted reproduction

# Low sperm to egg ratio required for successful *in vitro* fertilization in a pair-spawning teleost, Senegalese sole (*Solea senegalensis*)

Sandra Ramos-Júdez[1],[†], Wendy Ángela González-López[1],[†], Jhons Huayanay Ostos[1], Noemí Cota Mamani[2], Carlos Marrero Alemán[1], José Beirão[3] and Neil Duncan[1]

[1]IRTA, Sant Carles de la Ràpita, C. Poble Nou km. 5.5, 43540 Sant Carles de la Ràpita, Tarragona, Spain
[2]Dirección General de Investigaciones en Acuicultura, Instituto del Mar del Perú (IMARPE), Lima, Peru
[3]Faculty of Biosciences and Aquaculture, Nord University, 8049 Bodø, Norway

SR-J, 0000-0003-4242-5894; WÁG-L, 0000-0002-4378-9123;
JHO, 0000-0002-5983-0557; NCM, 0000-0001-8944-8159;
CMA, 0000-0002-5726-6710; JB, 0000-0002-4444-7563;
ND, 0000-0001-5456-9217

**Author for correspondence:**
Neil Duncan
e-mail: neil.duncan@irta.cat

[†]These two authors made a similar contribution to the article.

Cultured Senegalese sole (*Solea senegalensis*) breeders fail to spawn fertilized eggs. The implantation of large-scale *in vitro* fertilization protocols, to solve this problem, has been frustrated by low production of poor quality sperm. Cultured females were induced to ovulate with a $5 \, \mu g \, kg^{-1}$ single injection of gonadotropin-releasing hormone agonist (GnRHa) and viable eggs ($82.6 \pm 9.2\%$ fertilization) were stripped $41:57 \pm 1:46$ h after the injection. Sperm was collected from cultured males, diluted in modified Leibovitz and used fresh to fertilize the eggs. Males were not treated with hormones. A nonlinear regression, an exponential rise to a maximum ($R = 0.93$, $p < 0.0001$) described the number of motile spermatozoa required to fertilize a viable egg and 1617 motile spermatozoa were sufficient to fertilize $99 \pm 12\%$ ($\pm 95\%$ CI) of viable eggs. Similar, spermatozoa $egg^{-1}$ ratios of $592 \pm 611$ motile spermatozoa $egg^{-1}$ were used in large-scale *in vitro* fertilizations ($190\,512 \pm 38\,471$ eggs). The sperm from a single male ($145 \pm 50 \, \mu l$ or $8.0 \pm 6.8 \times 10^8$ spermatozoa) was used to fertilize the eggs. The mean hatching rate was $70 \pm 14\%$ to provide $131\,540 \pm 34\,448$ larvae per fertilization. The viability of unfertilized eggs stored at room temperature decreased gradually, and the sooner eggs were fertilized after

stripping, the higher the viability of the eggs. The collection of sperm directly into a syringe containing modified Leibovitz significantly increased the percentage of motile spermatozoa (33.4 ± 12.2%) compared with other collection methods. The spz egg$^{-1}$ ratios for Senegalese sole were at the lower end of ratios required for fish. Senegalese sole have a pair-spawning reproductive behaviour characterized by gamete fertilization in close proximity with no sperm competition. The provision of a large-scale *in vitro* fertilization protocol (200 µl of sperm per 100 ml of eggs) will enable the industry to operate sustainably and implement breeding programmes to improve production.

## 1. Introduction

Senegalese sole (*Solea senegalensis*) is a promising emerging aquaculture species in Europe. Sole production from land-based farms in Spain, Portugal, France and Iceland has increased rapidly to 1700 t in 2019 [1]. This increase is driven by good market prices, high market demand and successful culture practices [2] that permit cost-effective production despite the need for high levels of investment in culture infrastructure.

However, the production cycle is not fully controlled and relies on the capture of wild broodstock that spawn sufficient numbers of eggs to achieve targeted aquaculture productions [3,4]. The progeny of these wild broodstock and in particular the males exhibit a reproductive behavioural dysfunction and do not participate in the courtship to fertilize eggs [5–8]. Consequently cultured broodstocks that were reared entirely in captivity produce unfertilized eggs [5,7], as when the females release eggs into the water column males did not participate to fertilize the eggs [8]. Hormone-induced spontaneous spawning of cultured broodstocks had the same problem and only produced unfertilized eggs [9]. Differences between wild and cultured breeders have suggested that the reproductive dysfunction has a basis in a combination of endocrine reproductive control [9,10], social conditions during rearing [6,8], broodstock nutrition [11–15] and olfactory capacity [16]. However, no practical approach has been developed to overcome these reproductive dysfunctions and the low levels of fertilized eggs obtained from cultured broodstocks [6,9] were insufficient to meet industry needs.

In flatfish culture, *in vitro* fertilization methods are commonly used to obtain the fertilized eggs required for aquaculture [17]. *In vitro* fertilization enables aquaculturists to bypass behavioural reproductive dysfunctions of the type observed in cultured male Senegalese sole. However, the application of *in vitro* fertilization methods for Senegalese sole has been frustrated by the small quantities of poor quality sperm produced by males [18–22]. Although wild males compared with culture males produced slightly higher quality and quantities of sperm [21], both wild and cultured males produce small quantities of poor quality sperm [18–22]. Rasines *et al.* [23,24] has described *in vitro* fertilization procedures for Senegalese sole on an experimental scale. Cultured female sole were induced to ovulate with gonadotropin-releasing hormone agonist (GnRHa) and batches of 1 ml of stripped eggs were fertilized with 30 µl of cryopreserved sperm from cultured males. Similarly, Liu *et al.* [25] described *in vitro* fertilization of Senegalese sole eggs on an industrial scale. Again, eggs were obtained from GnRHa-induced cultured females and all the eggs from each female were fertilized *in vitro* with the sperm from three or four cultured males. However, few details were given on the amount of sperm or eggs used, and the spermatozoa (spz) to egg ratio was not detailed.

When sperm is limiting, it is of critical importance to know the spz to egg ratio (amount of spz required to fertilize each egg) to plan *in vitro* fertilization procedures. Spermatozoa to egg ratios for *in vitro* fertilization of fish eggs show considerable variation ranging from $\times 10^3$ to $\times 10^6$ [26]. However, it would appear that flatfish have lower spz requirements, as winter flounder (*Pseudopleuronectes americanus*) required $3.4 \times 10^4$ spz egg$^{-1}$ [27] and turbot (*Scophthalmus maximus*) required just 3000–6000 spz egg$^{-1}$ [28,29]. Many flatfish species have reproductive strategies and behaviours to spawn as a pair [30,31]. Monogamous fish species (classified as species that spawn in a pair) were shown to have smaller testes compared with polyandrous species (group spawning of males with a female) [32]. Monogamy or pair-spawning will also reduce sperm competition (sperm from two or more males compete to fertilize the eggs) and decreasing sperm competition has been related to smaller testes and lower sperm production [33,34]. Senegalese sole spawn as pairs [5,30] and spawning pairs show a degree of fidelity during and between spawning seasons [4,6]. Therefore, pair-spawning and low sperm competition would appear to explain the small testes size [22,35] and low sperm production reported for Senegalese sole [18–22]. In addition, a preliminary study found that Senegalese sole achieved a high percentage of fertilization with a low spz to egg ratio [36] and

indicated that *in vitro* fertilization with the low numbers of sperm may be a viable solution to the industry's problem to control the reproduction.

The present study aimed to determine the spz to egg ratio required for *in vitro* fertilization in Senegalese sole. The first aim was to determine the sperm to egg ratio on an experimental scale and then use similar ratios for commercial large-scale *in vitro* fertilizations as a proof of concept. Additional aims were, to determine the viability of ovulated eggs stored at room temperature and to improve sperm collection methods.

# 2. Material and methods

## 2.1. Experimental animals

All Senegalese sole broodstock used were cultured fish that had been hatched and reared entirely in captivity. Females used had an average weight of $1.53 \pm 0.28$ kg and males had a weight of $1.05 \pm 0.25$ kg. Fish were maintained in 10 000 l tanks in IRTA Sant Carles de la Rápita (Catalonia, Spain). Prior to experiments, fish were held in surface seawater (approx. 35 ppt, greater than 5 mg l$^{-1}$ O$_2$) and a controlled natural temperature cycle (9–20°C) using recirculation systems (IRTAmar). Tanks were covered with shade netting and photoperiod was natural with natural light. The fish were fed 4 days a week with either unfrozen polychaetes and mussels (0.75% of biomass) or 5 mm pelleted Broodfeedlean broodstock diet (0.55% of biomass) (Sparos, Olhão, Portugal). During experiments conducted from April to June 2019, fish were held in the same conditions with the exceptions that water temperature was maintained at a constant $16 \pm 1$°C and fish were not fed 24 h before any manipulation.

## 2.2. Gametes

Eggs were obtained by inducing ovulation. Ovarian biopsies were taken from females with swollen ovaries and the diameter of 20 oocytes were measured (×40 Carl Zeiss Axiostar microscope). Females were selected that had mean oocyte diameter greater than 600 µm [36]. The females were administered 5 µg kg$^{-1}$ of GnRHa (Sigma code L4513, Sigma, Spain) [37] between 18.00 and 19.00 h. The females were held with constant temperature ($16 \pm 1$°C) and total darkness until ovulation. Females were checked for ovulation every 2–3 h starting from 40 h after the administration of GnRHa [23,24,36] and all eggs were stripped from the ovulated females. Percentage fertilization when spz were in excess was used to indicate egg quality.

Sperm was obtained from males by repeatedly, gently massaging the testes and applying pressure along the full length of the sperm ducts to the urogenital pore. All sperm with urine contamination was collected in a 1 ml syringe. González-López *et al.* [22] demonstrated that avoiding urine contamination was almost impossible and that high numbers of motile spz were obtained stripping sperm mixed with urine. The volume collected was measured with the syringe to an accuracy of 10 µl and the sperm was transferred to a 1.5 ml Eppendorf and immediately diluted with modified Leibovitz [22] using the dilution required for the experiment (see below). The sperm motility was initially observed (×100 Carl Zeiss Axiostar microscope) by activating 1 µl of diluted sperm with 19 µl of clean seawater. Sperm samples with low or no motility were rejected. All sperm samples were stored over ice or at 4°C (refrigerated) until analysis or used to fertilize eggs.

The spz concentration (spz ml$^{-1}$) was measured using a Thoma cell counting chamber. A 10 µl sample of sperm was diluted 1 : 500 in 10% formalin and 10 µl of the dilution was pipetted into the counting chamber. After 10 min for spz to sediment, the chamber was observed using a microscope (×100 magnification with Olympus BH microscope), photographed (IC Capture software and GigE digital camera model: DMK 22BUC03 Monochrome, The Imaginsource, Bremen, Germany) and the number of spz counted (ImageJ software, http://imagej.nih.gov/ij/).

Sperm motility parameters were determined as described by González-López *et al.* [22]. Spermatozoa were activated by mixing 1 µl of diluted sperm (1 : 4 with modified Leibovitz) with 20 µl of seawater with 30% bovine serum albumin (BSA, Sigma, Spain). One microlitre of activated sperm was pipetted into an ISAS R2C10 counting chamber (Proiser R + D, S.L., Paterna, Spain) previously mounted and focused on the microscope (200× magnification Olympus BH). Video recording was initiated when spz were activated and tracks were recorded (IC Capture software and GigE digital camera) until motion ceased. Videos (AVI format) of spz tracks from 15 to 17 s (unless otherwise stated) after activation

were converted into image sequences (jpeg format using Virtual Dub 1.10.4 software http://www.virtualdub.org/). The image sequences were analysed using ImageJ software with the computer-assisted sperm analysis (CASA) plugin (ImageJ http://rsb.info.nih.gov/ij/plugins/) using the settings: brightness and contrast, −10 to 15/224 to 238; threshold, 0/198 to 202; minimum sperm size (pixels), 10; maximum sperm size (pixels), 400; minimum track length (frames), 10; maximum sperm velocity between frames (pixels), 30; frame rate, 30; microns/1000 pixels, 303; print motion, 1; the additional settings were not modified. The parameters, percentage of motile spz (% motility), curvilinear velocity (VCL, $\mu m\,s^{-1}$) and average path velocity (VAP, $\mu m\,s^{-1}$) were recorded. All sperm samples were analysed in triplicate.

The experiments (unless otherwise stated) aimed to use gametes (eggs and sperm) as soon as possible after collection to avoid possible loses of viability due to storage. Researchers worked as two groups to strip eggs and sperm at the same time and complete *in vitro* fertilizations soon afterwards.

## 2.3. Spermatozoa to egg ratio experiment

Five different females and five different males were used during this experiment. When an ovulated female was encountered, sperm was collected and checked to find a male with greater than 300 μl of sperm that exhibited motility. The sperm was serially diluted with modified Leibovitz [22] to achieve eight dilutions: 1 : 4; 1 : 19; 1 : 79; 1 : 319; 1 : 959; 1 : 2879; 1 : 5759; 1 : 11 519. A sample of the first dilution (1 : 4) was used to determine the spz concentration and percentage motility. The spz concentration in the dilution 1 : 4 was used to calculate the spz concentration in each dilution and spz motility to calculate the concentration of motile spz. The eggs and diluted sperm were used to make triplicate fertilizations for each serial dilution. Fertilizations were made in 100 ml beakers by pipetting in close sequence, 0.5 ml of eggs, 20 μl of diluted sperm and 5 ml clean seawater. A 1 ml pipette with a cut tip was used to pipette eggs and a 100 μl pipette with a cut tip was used to pipette diluted sperm. The eggs, sperm and seawater were gently mixed by rocking and swirling the beaker. After 3–5 min, the volume of seawater was topped up to 100 ml. The beakers of fertilized eggs were transferred to a 16°C incubator. After 24 h, the eggs from each beaker were concentrated in a sieve and placed in a 10 ml Bogorov chamber and greater than or equal to 50 eggs were randomly examined using a binocular microscope (Nikon C-DSS230) to determine the number of developing eggs. In addition, the number of eggs in three 0.5 ml samples was counted for each female.

## 2.4. Proof-of-concept experiment

Seven different females and seven different males were used during this experiment. The experiment aimed to make large-scale *in vitro* fertilizations using spz egg$^{-1}$ ratios from the previous experiment to fertilize the number of eggs (greater than 100 000 eggs) that would be required in a commercial fish farming scenario. When an ovulated female was encountered, sperm was collected and checked to find a male with greater than or equal to 150 μl of motile sperm. The sperm was immediately diluted 1 : 4 in modified Leibovitz and a sample of 50 μl of diluted sperm was taken to determine spz concentration and percentage motility (CASA). All the eggs were stripped from the female into a clean, dry 1 l jug and the volume of eggs was measured with an accuracy of 10 ml. Three samples of 0.5 ml of eggs were taken and counted. The remaining sperm obtained from the male was added to the eggs followed by a volume of seawater that was equal to the volume of eggs. The eggs, sperm and seawater were gently swirled to mix the contents. After 2–3 min the jug was topped up to 1 l with seawater. The eggs were placed in 30 l incubators with the same conditions as the broodstock holding tanks. The number of eggs in each incubator was estimated by mixing the incubator homogenously and taking three 100 ml samples and counting the eggs in each sample. The eggs were left 2 days to hatch and the number of hatched larvae in each incubator was estimated as above for the eggs. The hatch rate was calculated from the number of eggs stocked and number of larvae hatched for each female–male pair.

## 2.5. Egg viability experiment

Three different females and three different males were used during this experiment. When an ovulated female was encountered, males were checked to find a male with greater than or equal to 100 μl of motile sperm. The sperm was immediately diluted 1 : 4 in Leibovitz. All the eggs were stripped from the female into a clean, dry 1 l jug. The eggs were covered and stored at room temperature inside a building (out of

sunlight). As soon as possible after the gametes were stripped the first fertilization was completed as previously described. The time the eggs were stripped and the time the first fertilization was made was recorded. Further fertilizations were completed at 30–60 min intervals. Fertilizations were completed in duplicate or triplicate using 0.5 ml of eggs and 20 µl of diluted sperm that ensured an excess of motile spz per egg. As described above, the beakers of fertilized eggs were transferred to a 16°C incubator and after 24 h the percentage of developing eggs was determined for each fertilization.

## 2.6. Sperm collection experiment

Thirteen males were used for this experiment. Sperm from each male was collected as previously described, with the exception that the sperm was collected either into an empty clean syringe (100 µl of sperm) or into a syringe that contained modified Leibovitz (to give a 1 : 4 dilution, 50 µl of sperm collected into 200 µl of Leibovitz). Both collection methods were used for each male and the sequence of the collection was alternated. For seven animals sperm was first collected into a clean syringe and then into a syringe that contained modified Leibovitz and for six animals the reverse, first directly into modified Leibovitz and then a clean syringe. The 100 µl of sperm collected into a clean syringe was immediately divided into 50 µl that was diluted in 200 µl of Leibovitz (1 : 4 dilution) and 50 µl that was kept as undiluted sperm as a control. The time of collection and dilution after collection was recorded. The sperm motility parameters were analysed (CASA) for the samples collected by these three methods: collected directly into Leibovitz, collected before dilution in Leibovitz and undiluted sperm. Sperm motility parameters were analysed 30 s after activation for 2 s. The parameters were measured at the time of collection (0 h), and 6 and 24 h after collection.

## 2.7. Data analysis and statistics

All means are with one standard deviation unless otherwise stated. For the spermatozoa egg$^{-1}$ ratio experiment, the percentage of viable eggs fertilized was calculated by dividing the actual fertilization rate by the mean fertilization rate when sperm was in excess. The number of motile sperm was calculated by multiplying the volume of diluted sperm added by the spermatozoa concentration and the percentage motility. To examine the effect that gamete quality among the five pairs of fish had on fertilization, a linear regression was applied to percentage of motile sperm (sperm quality) against number of sperm (motile and immotile) required per viable egg and to percentage of viable eggs (egg quality) against number of motile sperm required per egg. To describe the variation of percentage of viable eggs fertilized in relation to the number of motile spz per egg, a nonlinear regression based on an equation for an exponential rise to a maximum with double, five parameters was applied to the data. For the egg viability experiment, the percentage of viable eggs fertilized was calculated as above. To describe the variation of percentage of viable eggs fertilized in relation to time the eggs were stored at room temperature a nonlinear regression based on an equation for a four-parameter logistic curve was applied to the data.

The dataset for percentage motility from the sperm collection experiment was not normally distributed with a high percentage of zeros and skewed positively to a few higher values. The dataset could not be transformed to normality. The data were analysed twice. In one analysis, the data from the time point zero were ranked and analysed with a two-way ANOVA with the independent variables order of collection (first or second) and collection treatment (collected directly into Leibovitz, diluted in Leibovitz and undiluted sperm). In a second analysis, the dataset was scored into samples with or without motility and a $\chi^2$ analysis was made to compare expected proportions of samples with motility with actual proportions with motility between sperm collection treatments (collected directly into Leibovitz, diluted in Leibovitz and undiluted sperm) and time of storage (0 h, 6 h and 24 h). The Marascuillo procedure [38] was used to make a multiple comparison between proportions of individual treatments and time points. All the samples with motility were then separated (i.e. all the zeros were excluded) into a smaller dataset that was transformed to normality with the Logit transformation. The transformed data were compared with a one-way ANOVA followed by Holm–Sidak pairwise multiple comparison to compare mean motility for each treatment at each time point. A $p < 0.05$ was used to indicate significant differences. All statistical comparisons and regressions were made using Sigma Plot 12 (Systat Software, Inc., San Jose, CA 95110, USA) with the exception of the Marascuillo procedure that was completed with an Excel (Microsoft) worksheet written by the authors.

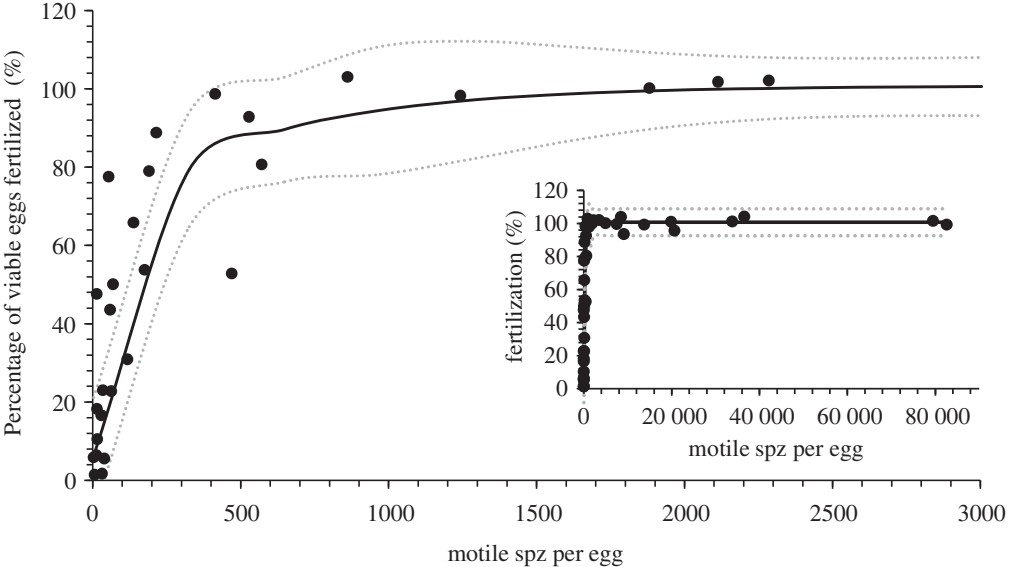

**Figure 1.** The percentage of viable eggs fertilized in relation to the number of motile spermatozoa (spz) per viable egg for Senegalese sole (*Solea senegalensis*). The insert figure shows the entire dataset up to over 80 000 spz per egg and the large figure shows a close up of the data up to 3000 motile spz per egg. The continuous line shows a nonlinear regression based on an equation for an exponential rise to a maximum with double, five parameters ($R = 0.93$, $p < 0.0001$) that represents the variation in percentage of viable eggs fertilized in relation to number of motile sperm per egg. The dotted lines indicate 95% confidence intervals for the nonlinear regression.

## 3. Results

A total of 46 cultured males were checked to obtain 28 (60.9%) males with the required sperm quantity and quality for the experiments. All males were only used once. A total of 20 cultured females were selected by ovarian swelling, and oocyte diameter determined. Five females were rejected as the ovaries contained ovulated ova and two as the ovaries had solid cysts. A total of 13 females had oocytes greater than or equal to 600 µm and were induced with GnRHa. Five females were not used as four females did not ovulate and one female had low quality eggs (0% fertilization). Eight (62%) females ovulated good quality eggs that were used in the different experiments. The eggs from some females were used for more than one experiment. The mean latency time from injection with GnRHa to the ovulation was 41:57 ± 1:46 h and mean fertilization was 82.6 ± 9.2%. The mean fecundity was 130 789 ± 36 723 eggs fish$^{-1}$ or 87 174 ± 24 378 egg kg$^{-1}$ of female body weight. The maximum time difference between stripping eggs and sperm was 30 min; therefore, either sperm or eggs were stored for 30 min or less before fertilization experiments were initiated.

### 3.1. Spermatozoa to egg ratio experiment

The percentage of viable eggs fertilized in relation to the number of motile spz per egg showed a rapid increase from zero that was represented by a nonlinear regression based on an equation for an exponential rise to a maximum with double, five parameters ($R = 0.93$, $p < 0.0001$) (figure 1). The nonlinear regression described that only 326 motile spz per egg fertilized 79 ± 15% (±95% confidence interval (CI)) of viable eggs, that 649 motile sperm fertilized 90 ± 13% (±95% CI) of viable eggs and 1617 motile spz fertilized 99 ± 12% (±95% CI) of viable eggs.

Among the five pairs, the percentage motility of the spz was correlated to the number of spz (motile and non-motile) required to fertilize a viable egg ($R^2 = 0.83$, $p = 0.021$). However, there was no correlation between percentage of viable eggs and number of motile sperm required to fertilize each egg ($R^2 = 0.03$, $p = 0.37$). Caution is required in the interpretation as $n$ was low ($n = 5$) and the statistical power (at $\alpha = 0.05$) of the tests was 0.66 and 0.13, respectively, for percentage motility and viable eggs.

## 3.2. Proof-of-concept experiment

The proof-of-concept large-scale *in vitro* fertilizations ($n = 7$) gave a mean percentage hatch of $70 \pm 14\%$ to produce a mean of $131\,540 \pm 34\,448$ larvae per fertilization (table 1). The sperm from a single selected cultured male with a volume of $145 \pm 50\,\mu l$ and total spz count of $8 \pm 6.8 \times 10^8$ was sufficient to fertilize large numbers of eggs ($190\,512 \pm 38\,471$) and produce large numbers of larvae ($131\,540 \pm 34\,448$). The mean number of spz per egg used for the commercial fertilizations was $2981 \pm 2932$ spz egg$^{-1}$ or $592 \pm 611$ motile spz egg$^{-1}$.

## 3.3. Egg viability experiment

The percentage of viable eggs fertilized decreased gradually after being stripped (figure 2). The nonlinear regression based on an equation for a four-parameter logistic curve ($R = 0.80$, $p = 0.008$) represented the variation in percentage of viable eggs fertilized in relation to the time eggs from the three females were stored at room temperature. The nonlinear regression indicated that after 30 min, the percentage fertilization had decreased from 100% to $81 \pm 26\%$ ($\pm 95\%$ CI), after an hour to $57 \pm 20\%$ ($\pm 95\%$ CI) and after 2 h to $32 \pm 19\%$ ($\pm 95\%$ CI) fertilization of viable eggs.

## 3.4. Sperm extraction experiment

The order of collection of the sperm appeared to have a significant effect and the first part of the sperm collected had significantly ($p = 0.04$) higher motility than the second part (figure 3). However, caution is required in the interpretation as there was considerable variation and the statistical power of the test was low (at $\alpha = 0.05$, power $= 0.44$). The percentage of samples that had motility decreased significantly ($p < 0.001$) with time (figure 4). At the time of collection ($t = 0$) there were no differences in the percentage of samples with motility, but after 6 and 24 h of storage at 4°C the samples collected directly into modified Leibovitz had significantly ($p < 0.05$) more samples with motility. The percentage motility was significantly higher ($p < 0.05$) in samples collected directly into modified Leibovitz at $t = 0$ compared to other time points ($t = 6$ and 24) and other collection methods, diluted in modified Leibovitz and undiluted (control), at all time points ($t = 0$, 6 and 24) (figure 5). The motility of samples collected directly into Leibovitz and stored for 6 h was significantly ($p < 0.05$) higher than undiluted sperm samples at collection ($t = 0$). The time difference in mixing sperm with Leibovitz was $4 \pm 2$ min, between samples collected directly into Leibovitz and the sperm diluted in Leibovitz after collection.

# 4. Discussion

## 4.1. Low spermatozoa to egg ratio in Senegalese sole

The present study demonstrated that low numbers of spermatozoa egg$^{-1}$ ensured high levels of fertilization in Senegalese sole. An exponential rise to a maximum ($R = 0.93$, $p < 0.0001$) described the number of motile spz required to fertilize a viable egg, the relationship rose steeply from 0 to 90% fertilization and from approximately 500 motile spz plateaued at 90–100% fertilization. A total of 1617 motile spz per egg were sufficient to fertilize $99 \pm 12\%$ ($\pm 95\%$ CI) of viable eggs. A low mean spz egg$^{-1}$ ratio also gave high rates of hatching in the proof-of-concept trials with large-scale *in vitro* fertilizations. A mean of $190\,512 \pm 38\,471$ eggs from a single female were fertilized with a mean volume of sperm of $145 \pm 50\,\mu l$ from a single male. This volume of sperm provided a mean of $592 \pm 611$ motile spz per viable egg, which was sufficient to achieve $70 \pm 14\%$ hatch and produce $131\,540 \pm 34\,448$ larvae per *in vitro* fertilization originating from single pair of fish. The spz egg$^{-1}$ ratios for Senegalese sole were at the lower end of ratios required for fish. Studies have shown that different fish species require a wide range of spz egg$^{-1}$ ratios to achieve high rates of fertilization [26]. These ratios ranged from 3000 spz egg$^{-1}$ for turbot [28] to $1 \times 10^5$ spz egg$^{-1}$ for Atlantic cod (*Gadus morhua*) [39]. Large-scale *in vitro* fertilization protocols used in a hatchery generally refer to the volume of sperm needed to fertilize eggs, for example in rainbow trout 1 ml of sperm was recommended for 10 000 eggs [40] and for Atlantic halibut (*Hippoglossus hippoglossus*) 1 ml of sperm for 1 l of eggs [41]. For Senegalese sole, the present study has shown that a conservative estimation for hatcheries based on the experiments and the 95% standard deviations, would be 200 µl of sperm (approx. $2.5 \times 10^8$ motile spz) to fertilize 100 ml of eggs (approx. 150 000 eggs).

**Table 1.** Data from seven commercial scale *in vitro* fertilizations made for Senegalese sole (*Solea senegalensis*). Latency time from application of GnRHa (5 µg kg$^{-1}$) to stripping of ovulated eggs, volume of eggs used, total number of eggs used, volume of sperm used (sperm from single male before dilution), total number of spermatozoa (spz) added, number of spz per egg added, percentage motility of spz, number of motile spz per egg, mean percentage hatch in incubators and total number of larvae produced.

| female | latency time | vol. eggs (ml) | total no. eggs | vol. sperm (µl) | total no. spz. × 10$^8$ | no. spz. per egg | percentage spz motility | total no. motile spz. per egg | percentage hatch ± s.d. | total no. larvae |
|---|---|---|---|---|---|---|---|---|---|---|
| 1 | 41:20 | 120 | 161 280 | 140 | 14.0 | 8681 | 10.4 | 905 | 82 ± 16 | 132 415 |
| 2 | 43:15 | 150 | 252 450 | 140 | 3.3 | 129 | 21.9 | 28 | 79 ± 14 | 198 530 |
| 3 | 40:45 | 75 | 144 250 | 100 | 6.9 | 4802 | 38.1 | 1830 | 78 ± 24 | 112 600 |
| 4 | 42:00 | 110 | 181 940 | 120 | 1.5 | 824 | 30.0 | 247 | 58 ± 10 | 104 625 |
| 5 | 39:30 | 100 | 198 467 | 160 | 3.5 | 1747 | 22.2 | 388 | 77 ± 7 | 153 245 |
| 6 | 43:40 | 100 | 167 500 | 120 | 4.9 | 2925 | 8.3 | 242 | 71 ± 11 | 117 500 |
| 7 | 44:40 | 150 | 227 700 | 120 | 4.0 | 1757 | 28.8 | 507 | 44 ± 9 | 101 865 |
| **mean ± s.d.** | **41:57** | **115** | **190 512** | **145** | **8.0** | **2981** | **22.8** | **592** | **70** | **131 540** |
|  | 01:46 | 27.5 | 38 471 | 50 | 6.8 | 2933 | 10.7 | 611 | 14 | 34 448 |

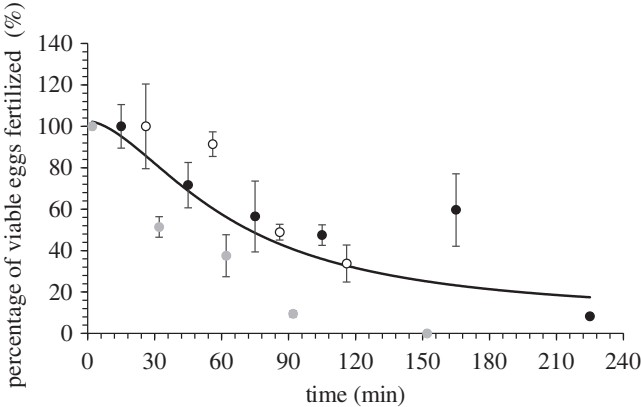

**Figure 2.** The percentage of viable eggs fertilized in relation to the time eggs were stored at room temperature for three Senegalese sole (*Solea senegalensis*) females. Different dots represent different females. The line shows a nonlinear regression based on an equation for a four-parameter logistic curve ($R = 0.08$, $p = 0.008$) that represents the variation in percentage of viable eggs fertilized in relation to time the eggs from the three females were stored.

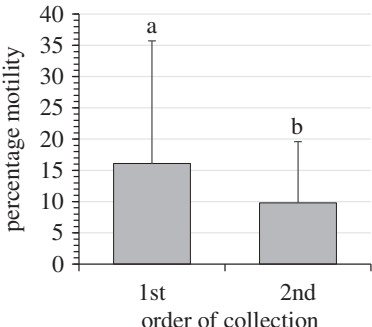

**Figure 3.** Mean percentage motility (±1 standard deviation) of sperm samples collected first or second from each Senegalese sole (*Solea senegalensis*) male ($n = 13$). Different letters indicate a significant difference ($p = 0.04$, $\alpha = 0.05$, power = 0.44).

## 4.2. Improved *in vitro* protocol

An aspect that complicates *in vitro* fertilization protocols is ensuring good gamete quality for fertilization. Generally, the sperm is obtained first and stored until eggs are obtained [42]. Considering that Senegalese sole sperm has poor quality [22] and eggs have a short period of viability [23,24], the storage of eggs and collection methods of sperm were examined in the present study to improve the *in vitro* fertilization protocol. In addition, the present study ensured sperm or eggs were not stored for longer than 30 min before fertilization to limit the effect of gamete quality deterioration. This proved to be necessary as the decline in quality of stripped Senegalese sole eggs stored at room temperature was gradual and continuous. Egg quality appeared to decline during the first 30 min of storage with no plateau period of good egg quality, which indicated the sooner eggs were fertilized after stripping the higher the viability of the eggs. This decline in egg quality after ovulation and stripping has been described in a wide range of species as the overripening process, where eggs age and in association with morphological and biochemical changes lose viability and fertilization rates decline [17,43,44]. A rapid decline in egg quality has been observed in other species, but the decline was initiated after a period of good egg quality of 1 h in curimata (*Prochilodus marggravii*) [45] and 50 min in meagre (*Argyrosomus regius*) [43]. There is considerable variation across species and some species have very different egg storage capacities, for example, eggs from a Cyprinidae species kutum (*Rutilus frisii*) [44] maintained good egg quality during 8 h of storage and salmonid eggs can be stored successfully for 4–5 days [40].

To improve sperm quality different methods of sperm collection were compared. The collection of sperm directly into modified Leibovitz significantly increased motility at the time of collection and the storage capacity in terms of motility and number of samples with motility. Senegalese sole sperm is difficult or impossible to collect without urine contamination [22]. The urine has negative effects on

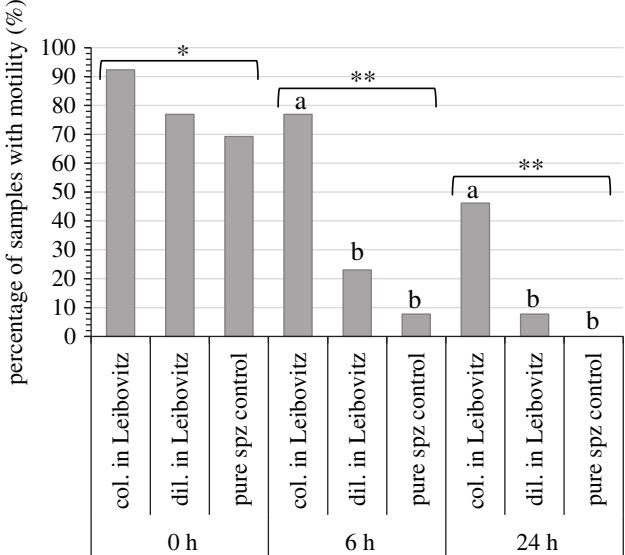

**Figure 4.** The percentage of sperm samples from Senegalese sole (*Solea senegalensis*) (*n* = 13) that had sperm motility for different time points 0 h after collection and after 6 and 24 h of storage at 4°C. Three sperm collection methods were tested, collected directly into modified Leibovitz (col. in Leibovitz), diluted in modified Leibovitz (dil. in Leibovitz) after collection and undiluted sperm (control). Different letters indicate a significant difference (*p* < 0.05) within a time point and different number of asterisks indicate significant difference (*p* < 0.05) between time points.

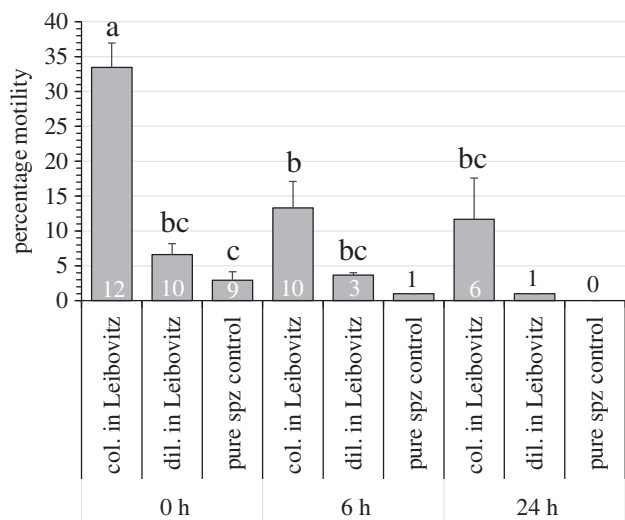

**Figure 5.** Mean percentage motility (±1 standard error of the mean) of sperm samples collected from Senegalese sole (*Solea senegalensis*) using three collection methods, collected directly into modified Leibovitz (col. in Leibovitz), diluted in modified Leibovitz (dil. in Leibovitz) after collection and undiluted sperm (control). The motility of samples was tested at different time points 0 h after collection and after 6 and 24 h of storage at 4°C. Different letters indicate a significant difference (*p* < 0.05) among collection methods and time points. The '*n*' of each mean is at the bottom of each column. Means with less than *n* = 3 were not included in the statistical test.

the sperm quality, which as in other species appeared to change osmolality and pH, which prematurely activated spz [22,46–48]. The collection into modified Leibovitz mitigated the negative effect of urine contamination to improve the sperm quality [22]. Other studies on species where urine contamination was difficult to avoid have focused on improved collection methods that reduce urine contamination with the use of catheters that were inserted into the sperm duct [49,50] or used a collecting pipette in combination with vacuum aspiration [51]. The very small volumes and small diameter of the urogenital pore in Senegalese sole make these kinds of approaches difficult or impossible. Small volumes of contaminated sperm obtained from some species of birds were collected directly into

syringes with extender to maintain sperm quality (IG Nebot, Rara Avis Biotec S.L., Valencia, Spain, personal communication). Therefore, in the present study, this approach was taken to reduce the time that spz were in contact with urine contamination before dilution in the Leibovitz extender. A significant improvement in sperm quality was obtained by collecting the sperm directly into a syringe containing modified Leibovitz, which reduced by $4 \pm 2$ min the time before sperm was diluted with Leibovitz.

## 4.3. Sustainable Senegalese sole aquaculture

The success of massive *in vitro* fertilizations, combined with indications of egg viability during storage and improved methods for collecting and managing sperm, provided a protocol that can be used on an industrial scale to obtain eggs from cultured breeders for hatchery production. At present, the Senegalese sole aquaculture industry relies on obtaining eggs from wild adult breeders captured in the commercial fishery and there is no sustainable fishery for Senegalese sole (https://fisheries.msc. org). Therefore, obtaining viable eggs from cultured breeders has been a bottleneck that makes the industry unsustainable and unable to implement breeding programmes. Breeding programmes are an essential part of an aquaculture business plan that enable companies to improve growth and product quality. However, ideally, reproduction must be controlled to enable the selection and production of viable gametes from any animal that has the desired production traits. In the present study, 61% of the males checked for sperm had the required quantity and quality needed for the experiments and *in vitro* fertilizations. This availability of males, has already been improved as therapies with recombinant gonadotropins exist that both increase sperm production and quality [52,53]. These recombinant gonadotropin therapies, significantly increased sperm production by up to seven times and significantly increased the sperm quality parameters, percentage motility, progressivity and velocity of spz.

In the present study, females were initially selected by ovarian swelling and it is unclear what percentage of females would be available for selection over a reproductive season. Of the females that were GnRHa induced, 62% ovulated good quality eggs. Therefore, studies are needed to identify the number of females available for GnRHa induction and to improve the success rate of GnRHa inductions. The present study used a GnRHa dose of $5 \, \mu g \, kg^{-1}$ to induce the ovulation of eggs compared with $25 \, \mu g \, kg^{-1}$ used by Rasines *et al.* [23,24]. These studies had similar holding conditions and temperature (16°C) and obtained very similar timing of ovulation with means close to 42 h (range of 39–44 h). Egg quality appeared to be higher in the present study, but differences in methods and particular sperm storage and usage make comparisons inappropriate. Different doses of GnRHa have been compared with induce spontaneous liberations of eggs in Senegalese sole. Agulleiro *et al.* [37] tested the injection of GnRHa doses of 1, 5 and $30 \, \mu g \, kg^{-1}$. The dose of $5 \, \mu g \, kg^{-1}$ produced the most eggs and the dose of $30 \, \mu g \, kg^{-1}$ produced no liberations of eggs. Guzmán *et al.* [54] compared injections of 5 and $25 \, \mu g \, kg^{-1}$ of GnRHa, but found no differences in number of eggs released between GnRHa injected and untreated control fish. However, the number of oocytes advancing to hydration in the females treated with $5 \, \mu g \, kg^{-1}$ appeared to be higher than in females treated with $25 \, \mu g \, kg^{-1}$ and controls. The present study, combined with studies on GnRHa-induced spontaneous liberations of eggs, would indicate that the lower dose of $5 \, \mu g \, kg^{-1}$ of GnRHa provided similar ovulation timing and egg quality as $25 \, \mu g \, kg^{-1}$ and perhaps suggest that better results may be obtained with lower doses.

## 4.4. Gamete quality and fertilization

Logically, the spz egg$^{-1}$ ratio required is related to the success of individual sperm to fertilize an egg. The success of spz will depend on factors of the fertilization environment and gamete quality/characteristics that hinder or aid the spz to reach the micropyle of the egg. The environment used for *in vitro* fertilization has been shown to affect the spz egg$^{-1}$ ratio. For example, the volume or space provided for fertilization affected the spz egg$^{-1}$ ratio, as larger volumes increased the space to be travelled to fertilize the egg and increased the number of spz required [28,55,56]. Therefore, variation in the fertilization environment complicates the comparison of different studies within and among species. However, as in the present study the fertilization environment can be standardized between tests and replicated to ensure results can be compared.

Gamete characteristics and/or quality vary among species and individuals. Variations in gamete quality have been shown to affect the spz egg$^{-1}$ ratio. Obviously, percentage sperm motility affects the

ratio [57,58], but velocity has also been shown to affect the ratio in walleye (*Sander vitreus*) [59] and pufferfish (*Takifugu niphobles*) [57]. Sperm with higher percentage motility [57,58] and spermatozoa with higher mean swimming speeds had lower spz egg$^{-1}$ ratios to fertilize a high percentage of eggs [57,59]. In the present study, sperm percentage motility was related positively with spz egg$^{-1}$ ratio ($R^2 = 0.83$, $p = 0.021$). Senegalese sole sperm has variable and generally poor quality with low percentage motility [22]. In the present study as in other studies [43], the effect that percentage motility has on the spz egg$^{-1}$ ratio was removed by examining the relationship between the number of motile sperm and eggs fertilized. Many studies simply express total number of sperm (including immotile sperm) in the spz egg$^{-1}$ ratio, however, this is inaccurate and should be stated with the percentage motility of the sperm used. Although the practice of using total spz is accepted in the literature it should only be applied to species that have little variation in motility and preferably high levels (close to 100%) of motility among individuals.

In the present study, the quality of eggs did not appear to be related to the spz egg$^{-1}$ ratio. However, the low variation in egg quality ($82.6 \pm 9.2\%$) and $n$ ($n = 5$) may have reduced the possibility to determine a relationship. Previous studies are contradictory indicating that eggs with the higher quality required more [43] and less [55] spz than low quality eggs. It would appear probable that both the quality and the characteristics of the unfertilized egg are implicated in the fertilization success. Good quality eggs would have more eggs to be fertilized and low quality eggs would have an environment with more space to encounter viable eggs among the inviable eggs. In addition, fish eggs of some species have been shown to have no mechanisms to attract spz while other species eggs have chemical and physical properties that guide the spz to the micropyle [60]. The ability to attract spz to the eggs would in theory reduce the spz egg$^{-1}$ ratio. These observations would suggest number of motile spz per viable egg should be used to ensure egg viability does not affect the spz egg$^{-1}$ ratio. However, further work is required to determine the effect of egg quality on the spz egg$^{-1}$ ratio.

The fertilization rates in the present study also indicated that cultured breeders did have the potential to provide good quality gametes. This agrees with observations that although wild males compared with culture males produced slightly higher quality and quantities of sperm [21], both wild and cultured males produce similar small quantities of poor quality sperm [18–22] and that cultured females held with wild males spawned similar quantities of eggs with similar quality as broodstocks of only wild fish [8].

## 4.5. Why do Senegalese sole have a low spz egg$^{-1}$ ratio?

Senegalese sole spawn as a female and male pair with no involvement of other individuals [30] and spawning pairs show a degree fidelity during and between spawning seasons [4,6]. Therefore, Senegalese sole fertilization does not involve sperm competition as all the sperm originates from a single male and does not compete with sperm from other males. In addition, the two sexes swim in synchrony with the genital pores held close together [30]. The male urogenital duct is slightly raised and the female oviduct forms a kind of well when eggs are being stripped (personal observation by the authors during the present study), which together with the closeness of the fish during gamete liberation [30] suggest that the male and female place the spz next to the eggs in very close proximity. Other studies have demonstrated that these behavioural and reproductive strategies are related to low spz egg$^{-1}$ ratios or low sperm production. Reducing the space of the fertilization environment has been shown to reduce the spz egg$^{-1}$ ratios required [55]. Different species have very varied strategies and behaviours that will alter the space of the fertilization environment, which can range from mass spawning in aggregations in open water [61,62] to spawning between two fish in an enclosed space or in very close proximity [30,63]. The spawning behaviour and number of individuals involved will influence the degree of sperm competition that gametes must negotiate to achieve fertilization. Sperm competition has been shown to influence fertilization success and the number and quality of spz that a species produces [33,34,64]. Monogamy and the absence of sperm competition were demonstrated to reduce testes size across different taxa, and monogamous fish species defined as spawning in a pair had significantly smaller testes compared with polyandrous species (group spawning of males with a female) [32]. Therefore, the Senegalese sole is a species that has been classified as pair-spawning (monogamous) that spawn in close proximity [30] and has been described to have low spz production [22,35] as well as a low spz egg$^{-1}$ ratio for fertilization (present study). The authors would suggest an interesting future investigation would be to explore and test the hypothesis that spz egg$^{-1}$ ratio is related to reproductive strategies and sperm production across different fish species including a range of flatfish species.

# 5. Conclusion

In conclusion, Senegalese sole require a low spz egg$^{-1}$ ratio to achieve high percentages of fertilization both on an experimental scale and in proof-of-concept large-scale *in vitro* fertilizations. The spz egg$^{-1}$ ratios for Senegalese sole, which exhibit pair-spawning in close proximity, were at the lower end of ratios required for fish. The protocol (200 µl of sperm per 100 ml of eggs) described in the present study will enable the Senegalese sole aquaculture industry to operate sustainably and establish breeding programmes.

Ethics. The fish were handled (routine husbandry and experimentation) in accordance with European regulations on animal welfare (Federation of Laboratory Animal Science Associations, FELASA, http://www.felasa.eu/). For all handling and sampling, fish were anaesthetized with 60 mg l$^{-1}$ tricaine methanesulfonate (MS-222; Sigma-Aldrich, Spain). The use of the fish, the experimental conditions and the procedures used were approved by the IRTA Ethics Committee with approval reference no. 7307.

Data accessibility. The dataset has been deposited at the Royal Society's figshare portal and is publicly available https://figshare.com/s/f535d4b147928673739b.

Authors' contribution. All authors made substantial contributions to conception and design and acquisition of data. Authors, S.R.-J., W.Á.G.L., J.H.O., J.B. and N.D. contributed to analysis and interpretation of data. Authors, S.R.J., W.Á.G.L. and N.D. drafted the article. S.R.-J., W.Á.G.L., J.B. and N.D. revised the article critically for important intellectual content. All authors gave final approval of the version to be published and agreed to be accountable for all aspects of the work in ensuring that questions related to the accuracy or integrity of any part of the work are appropriately investigated and resolved.

Competing interests. We declare we have no competing interests.

Funding. This study has been supported with funding from the Spanish National Institute for Agronomic Research (Instituto Nacional de Investigación y Tecnología Agraria y Alimentación—INIA)-European Fund for Economic and Regional Development (FEDER) (grant no. RTA2014-0048) coordinated by N.D. The study was also supported by the project 038433_REARLING, funded by Portugal and the European Union through ERDF, COMPETE 2020, in the framework of Portugal 2020, coordinated by Isidro Blanquet (Sea8 Group, Povoa de Varzim, Portugal) and in collaboration with Joan Cerdá (IRTA) and Ignacio Giménez, (Rara Avis Biotec S.L., Valencia, Spain). Participation by W.Á.G.-L was funded by a PhD grant from the National Board of Science and Technology (CONACYT, Mexico) and S.R.-J. by a PhD grant from AGAUR (Government of Catalonia) co-financed by the European Social Fund.

Acknowledgements. The authors would like to thank Josep Lluis Celades and the IRTA staff for technical assistance. Special thanks are also given for the participation and great enthusiasm to complete work to a high standard shown by Mario Villalta Vega and Alex Rullo Reverté, work experience students from the IES Alfacs Escola d'Aqüicultura (http://aquicultura.insalfacs.cat/). In particular, special thanks are given to Ignacio Giménez who refused an authorship after selflessly giving his advice and helping to improve sperm collection and storage. Lastly, we acknowledge that the research was driven by needs of the REARLING project (see funding statement) and discussions among the authors, Ignacio Giménez, (Rara Avis Biotec S.L., Valencia, Spain), Isidro Blanquet (Sea8 Group, Povoa de Varzim, Portugal) and Joan Cerdá (IRTA).

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
