## [Peer Review File · Royal Society Open Science]

Review History

RSOS-201718.R0 (Original submission)

Review form: Reviewer 1

Is the manuscript scientifically sound in its present form?

Yes

Are the interpretations and conclusions justified by the results?

Yes

Is the language acceptable?

Yes

Do you have any ethical concerns with this paper?

No

Have you any concerns about statistical analyses in this paper?

No

Recommendation?

Accept with minor revision (please list in comments)

Comments to the Author(s)

I found manuscript entitled „Low sperm to egg ratio required for successful in vitro fertilisation in a pair-spawning teleost, Senegalese sole (*Solea senegalensis*)“ as significant contribution in the field of applied research. The value of information it brings is high as commercial fishing of most of the fish species harvested is not sustainable. Fish aquaculture has its difficulties but is generally considered as much better solution than unsustainable overfishing what is unfortunately very common in recent days. Papers like the present one improving methodology of cultured fish species are thus really in demand.

Manuscript is generally well written and clear, but I still have some minor points which should be considered.

Line 9 I would rather use word “viable” than “good quality eggs”

Line 10 The point that males were not hormonally stimulated should be also mentioned.

Line 34 add „number” between “sufficient” and “eggs”

Line 35-42 Does it mean that fish are let to spawn on their own in captivity? Is it natural spawning? Not induced? It is not entirely clear, how progeny of wild caught parents produce unfertilized eggs. Females just release eggs without males involved?

Line 45-47 Does it mean than wild caught males have much higher quality of sperm than captive individuals? Please make this statement clear.

Line 86 add year when your experiments were conducted

Line 99 I am just wondering here, why the authors did not use hormonal stimulation in males as well. This is common practice as far as I have experience with and it works. The reason for not using hormonal stimulation of males could be mentioned in discussion.

Line 227 add „cultured” before males

Lines 242-245 I would appreciate some graphical illustration of this result

Line 258 4.3.

Line 293 add “originating from single pair of fish” after IVF.

Line 339 in this part of discussion I would appreciate some kind of comparison of quality and quantity between eggs and sperm of wild caught fish and captured ones. I really miss this in the manuscript.

Review form: Reviewer 2

Is the manuscript scientifically sound in its present form?

Yes

Are the interpretations and conclusions justified by the results?

Yes

Is the language acceptable?

Yes

Do you have any ethical concerns with this paper?

No

Have you any concerns about statistical analyses in this paper?

Yes

Recommendation?

Accept with minor revision (please list in comments)

Comments to the Author(s)

This study shows large scale of in vitro fertilization of the flatfish Senegalese sole. The paper is very well written and easy to understand. In terms of aquaculture, this report is useful for the practical protocol in vitro fertilization of the Senegalese sole. I like this paper and should be published in this journal after minor revision.

(However, the mating system is not important this paper and the quality of the study is not striking in terms of mating system described in the abstract.)

In table 1

Percentage of sperm motility is very low comparing with Fig 4. What is different between table 1 and fig 4?

Figure 1

The plotted circles were obtained same combinations or mixture of different combinations? If different combinations (like in fig2), is there any difference among combinations?

Figure 3

Please indicate number of individuals you tested (N= ? Males).

Figure 5

Position of "percentage motility" should be centered.

Numbers indicated on the bars are slipped off from the bars.

For example, "10" in Dil in Leibovitz

Error bars indicate S.D.?

The sperm motility of fig 5 is also very low like in table 1.

The following is my interest and please ignore if authors are not interested in. The discussion about monogamous and polygamous is weak to conclude monogamous species need smaller number of motile sperm. Because the authors did not investigate sperm motility and sperm number for fertilization success in the polygamous flatfish/flounder species (i.e., the wide-eyed flounder). The referenced the paper is not related species of Senegalese sole.

If authors investigate same view points using the flat fish species (>20 species) with different mating systems (monogamous or polygamous), and phylogenetically comparisons are conducted (such as PGLS), their discussion will be clearly shown in phylogenetically and statistically.

Moreover, I myself experienced many monogamous species did not produce large number of sperm and often faced smaller number of sperm with low percentage of motility. I really appreciate authors effort to conduct fertilization trials of this species.

Decision letter (RSOS-201718.R0)

Dear Dr Duncan

On behalf of the Editors, we are pleased to inform you that your Manuscript RSOS-201718 "Low sperm to egg ratio required for successful in vitro fertilisation in a pair-spawning teleost, Senegalese sole (*Solea senegalensis*)" has been accepted for publication in Royal Society Open Science subject to minor revision in accordance with the referees' reports. Please find the referees' comments along with any feedback from the Editors below my signature.

Please submit your revised manuscript and required files (see below) no later than 7 days from today's (ie 30-Nov-2020) date. Note: the ScholarOne system will 'lock' if submission of the revision is attempted 7 or more days after the deadline. If you do not think you will be able to meet this deadline please contact the editorial office immediately.

on behalf of Dr Kristina Sefc (Associate Editor) and Kevin Padian (Subject Editor)
openscience@royalsociety.org

Associate Editor Comments to Author (Dr Kristina Sefc):

Associate Editor: 1

Comments to the Author:

The two reviewers agree that the manuscript requires only small changes to make it suitable for publication. Please attend to their points. Please also consider the question about the support for your suggestion that monogamous species need a smaller number of motile sperm.

I am looking forward to receiving a revised version of your manuscript.

Reviewer comments to Author:

Reviewer: 1

Comments to the Author(s)

I found manuscript entitled „Low sperm to egg ratio required for successful in vitro fertilisation in a pair-spawning teleost, Senegalese sole (*Solea senegalensis*)“ as significant contribution in the field of applied research. The value of information it brings is high as commercial fishing of most of the fish species harvested is not sustainable. Fish aquaculture has its difficulties but is generally considered as much better solution than unsustainable overfishing what is

unfortunately very common in recent days. Papers like the present one improving methodology of cultured fish species are thus really in demand.

Manuscript is generally well written and clear, but I still have some minor points which should be considered.

Line 9 I would rather use word "viable" than "good quality eggs"

Line 10 The point that males were not hormonally stimulated should be also mentioned.

Line 34 add „number“ between „sufficient“ and „eggs“

Line 35-42 Does it mean that fish are let to spawn on their own in captivity? Is it natural spawning? Not induced? It is not entirely clear, how progeny of wild caught parents produce unfertilized eggs. Females just release eggs without males involved?

Line 45-47 Does it mean than wild caught males have much higher quality of sperm than captive individuals? Please make this statement clear.

Line 86 add year when your experiments were conducted

Line 99 I am just wondering here, why the authors did not use hormonal stimulation in males as well. This is common practice as far as I have experience with and it works. The reason for not using hormonal stimulation of males could be mentioned in discussion.

Line 227 add „cultured“ before males

Lines 242-245 I would appreciate some graphical illustration of this result

Line 258 4.3.

Line 293 add "originating from single pair of fish" after IVF.

Line 339 in this part of discussion I would appreciate some kind of comparison of quality and quantity between eggs and sperm of wild caught fish and captured ones. I really miss this in the manuscript.

Reviewer: 2

Comments to the Author(s)

This study shows large scale of in vitro fertilization of the flatfish Senegalese sole. The paper is very well written and easy to understand. In terms of aquaculture, this report is useful for the practical protocol in vitro fertilization of the Senegale sole. I like this paper and should be published in this journal after minor revision.

(However, the mating system is not important this paper and the quality of the study is not striking in terms of mating system described in the abstract.)

In table 1

Percentage of sperm motility is very low comparing with Fig 4. What is different between table 1 and fig 4?

Figure 1

The plotted circles were obtained same combinations or mixture of different combinations? If different combinations (like in fig2), is there any difference among combinations?

Figure 3

Please indicate number of individuals you tested (N= ? Males).

Figure 5

Position of "percentage motility" should be centered.

Numbers indicated on the bars are slipped off from the bars.

For example, "10" in Dil in Leibovitz

Error bars indicate S.D.?

The sperm motility of fig 5 is also very low like in table 1.

The following is my interest and please ignore if authors are not interested in. The discussion about monogamous and polygamous is weak to conclude monogamous species need smaller number of motile sperm. Because the authors did not investigate sperm motility and sperm number for fertilization success in the polygamous flatfish/flounder species (i.e., the wide-eyed flounder). The referenced the paper is not related species of Senegalese sole.

If authors investigate same view points using the flat fish species (>20 species) with different mating systems (monogamous or polygamous), and phylogenetically comparisons are conducted (such as PGLS), their discussion will be clearly shown in phylogenetically and statistically.

Moreover, I myself experienced many monogamous species did not produce large number of sperm and often faced smaller number of sperm with low percentage of motility. I really appreciate authors effort to conduct fertilization trials of this species.

===PREPARING YOUR MANUSCRIPT===

===PREPARING YOUR REVISION IN SCHOLARONE===

Author's Response to Decision Letter for (RSOS-201718.R0)

See Appendix A.

Decision letter (RSOS-201718.R1)

Dear Dr Duncan,

It is a pleasure to accept your manuscript entitled "Low sperm to egg ratio required for successful in vitro fertilisation in a pair-spawning teleost, Senegalese sole (*Solea senegalensis*)" in its current form for publication in Royal Society Open Science.

You can expect to receive a proof of your article in the near future. Please contact the editorial office (openscience@royalsociety.org) and the production office (openscience_proofs@royalsociety.org) to let us know if you are likely to be away from e-mail contact – if you are going to be away, please nominate a co-author (if available) to manage the proofing process, and ensure they are copied into your email to the journal.

on behalf of Dr Kristina Sefc (Associate Editor) and Kevin Padian (Subject Editor)
openscience@royalsociety.org

Appendix A

Crta. De Poble Nou, km 5.5
E-43540 Sant Carles de la Ràpita
(Tarragona, ESPAÑA)

Tel +34 977745427
Fax +34 977744138

Sant Carles de la Ràpita, 24th September 2020

Dear Editor,

We are pleased to re-submit to the journal ROYAL SOCIETY OPEN SCIENCE the manuscript entitled “**Low sperm to egg ratio required for successful *in vitro* fertilisation in a pair-spawning teleost, Senegalese sole (*Solea senegalensis*).**” to be considered for publication as a Research Article. Please see below that we have answered all the questions and suggestions from the reviewers. We take the opportunity to thank the Editor and Reviewers for helping improve the manuscript.

The manuscript details research to determine the number of spermatozoa required per egg for *in vitro* fertilisation procedures for the flatfish Senegalese sole (*Solea senegalensis*). We have shown that very low numbers of spermatozoa per egg were required, the lowest reported for a fish species. Senegalese sole males produce low numbers of poor quality sperm which has frustrated attempts to apply *in vitro* fertilisation procedures. The present study, will enable the emerging sole culture industry to use *in vitro* procedures to initiate sustainable culture, without the need to capture wild fish. The reasons for the low spermatozoan requirements are discussed and it is suggested hypothesis be tested to examine if low spermatozoa per egg ratios are related to reproductive strategies and thus the article orientates future research to an interesting aspect that may drive evolution of testes size and sperm production. Altogether, the present manuscript encourages further research and makes an important contribution to this area of research to draw citations and make a good contribution to Royal Society Open Science.

The authors have personal aims to publish in high quality, open access journals that are not driven by profits such as Royal Society Open Science. Although, our research article is well suited to journal on Aquaculture or applied science, we have not been able to find high quality, open access Aquaculture journals that are not driven by profits. For these reasons we would like that you consider this research article for publication in Royal Society Open Science.

We are looking forward to hearing from the Editorial board soon.

Yours faithfully,

Neil Duncan
IRTA Senior Researcher

Associate Editor Comments to Author (Dr Kristina Sefc):

Associate Editor: 1

Comments to the Author:

The two reviewers agree that the manuscript requires only small changes to make it suitable for publication. Please attend to their points. Please also consider the question about the support for your suggestion that monogamous species need a smaller number of motile sperm.

I am looking forward to receiving a revised version of your manuscript.

Authors reply: Thank you to you and the reviewers. We have answered all the changes requested as indicate below, as "Authors reply" after each point suggested by the reviewers.

Reviewer comments to Author:

Reviewer: 1

Comments to the Author(s)

I found manuscript entitled „Low sperm to egg ratio required for successful in vitro fertilisation in a pair-spawning teleost, Senegalese sole (*Solea senegalensis*)“ as significant contribution in the field of applied research. The value of information it brings is high as commercial fishing of most of the fish species harvested is not sustainable. Fish aquaculture has its difficulties but is generally considered as much better solution than unsustainable overfishing what is unfortunately very common in recent days. Papers like the present one improving methodology of cultured fish species are thus really in demand.

Manuscript is generally well written and clear, but I still have some minor points which should be considered.

Line 9 I would rather use word "viable" than "good quality eggs"

Authors reply: Words changed.

Line 10 The point that males were not hormonally stimulated should be also mentioned.

Authors reply: included the sentence: "Males were not treated with hormones."

Line 34 add „number" between "sufficient" and "eggs"

Authors reply: Included "numbers of"

Line 35-42 Does it mean that fish are let to spawn on their own in captivity? Is it natural spawning? Not induced? It is not entirely clear, how progeny of wild caught parents produce unfertilized eggs. Females just release eggs without males involved?

Authors reply: Yes that is what we mean. Text has been added to make this clear. "as when the females release eggs into the water column males did not participate to fertilise the eggs [8]. Hormone induced spontaneous spawning of cultured broodstocks had the same problem and only produced unfertilised eggs [9]."

Line 45-47 Does it mean than wild caught males have much higher quality of sperm than captive individuals? Please make this statement clear.

Authors reply: No it does not mean wild males produce higher quality sperm and although one study has shown small differences, it is debatable. To some extent it is irrelevant as fish from both origins produce small amounts of poor quality sperm. A sentence has been included on this point. "Although wild males compared to culture males produced slightly higher quality and quantities of sperm [21], both wild and cultured males produce small quantities of poor quality sperm [18–22]."

Line 86 add year when your experiments were conducted

Authors reply: Year added, 2019.

Line 99 I am just wondering here, why the authors did not use hormonal stimulation in males as well. This is common practice as far as I have experience with and it works. The reason for not using hormonal stimulation of males could be mentioned in discussion.

Authors reply: Hormone stimulation with the usual suspects (GnRHa and hCG) has been shown to have marginal increases in sperm production. New recombinant gonadotropin treatments have been very successful, but were not available for this study. This is mentioned in the discussion.

Line 227 add „cultured“ before males

Authors reply: “Cultured” has been included.

Lines 242-245 I would appreciate some graphical illustration of this result

Authors reply: Figure 1 is a graphical illustration of this result. The sentence is describing Figure 1.

Line 258 4.3.

Authors reply: Numbering corrected.

Line 293 add “originating from single pair of fish” after IVF.

Authors reply: “originating from single pair of fish” was included.

Line 339 in this part of discussion I would appreciate some kind of comparison of quality and quantity between eggs and sperm of wild caught fish and captured ones. I really miss this in the manuscript.

Authors reply: I think this is a little outside of the discussion of the manuscript. However, we have included a short text.

Reviewer: 2

Comments to the Author(s)

This study shows large scale of in vitro fertilization of the flatfish Senegalese sole. The paper is very well written and easy to understand. In terms of aquaculture, this report is useful for the practical protocol in vitro fertilization of the Senegale sole. I like this paper and should be published in this journal after minor revision.

(However, the mating system is not important this paper and the quality of the study is not striking in terms of mating system described in the abstract.)

In table 1

Percentage of sperm motility is very low comparing with Fig 4. What is different between table 1 and fig 4?

Authors reply: These are very different percentages. Table 1 shows the percentage of sperm that had motility. Figure 4 shows the percentage of sperm samples that had any degree, however, small of motility.

Figure 1

The plotted circles were obtained same combinations or mixture of different combinations? If different combinations (like in fig2), is there any difference among combinations?
Authors reply: The results from each male and female combination were plotted separately as in Fig. 2. Each point in figure 1 represents a fertilisation (made in triplicate) of eggs with a different concentration of sperm from a particular male + female combination.

Figure 3

Please indicate number of individuals you tested (N= ? Males).

Authors reply: Number of males was 13, now included en legend.

Figure 5

Position of "percentage motility" should be centered.

Numbers indicated on the bars are slipped off from the bars.

For example, "10" in Dil in Leibovitz

Authors reply: This appears to be a problem in the generation of the PDF as it is ok in the word document that was uploaded. Not sure what we can do about that?

Error bars indicate S.D.?

Authors reply: Error bars are SEM, indicated in legend.

The sperm motility of fig 5 is also very low like in table 1.

Authors reply: Yes sperm motility was low from all males and should not be confused with percentage of samples that had motility (see comment above).

The following is my interest and please ignore if authors are not interested in. The discussion about monogamous and polygamous is weak to conclude monogamous species need smaller number of motile sperm. Because the authors did not investigate sperm motility and sperm number for fertilization success in the polygamous flatfish/flounder species (i.e., the wide-eyed flounder). The referenced the paper is not related species of Senegalese sole.

If authors investigate same view points using the flat fish species (>20 species) with different mating systems (monogamous or polygamous), and phylogenetically comparisons are conducted (such as PGLS), their discussion will be clearly shown in phylogenetically and statistically.

Moreover, I myself experienced many monogamous species did not produce large number of sperm and often faced smaller number of sperm with low percentage of motility. I really appreciate authors effort to conduct fertilization trials of this species.

Authors reply: Our intention was not to conclude monogamous species need smaller number of motile sperm. This was the conclusion across other species (ref), which we cited.

The statements we had made on sole and monogamy were:

The low spz egg⁻¹ ratio required to fertilise all viable eggs was consistent with the reproductive behaviour and strategies of the species.

Consequently, the reproductive strategy and behaviour of Senegalese sole support described low spz production [22,35] as well as the low spz egg⁻¹ ratio for fertilisation (present study).

These statements were not only in reference to paired spawning (monogamy), but also to the absence of sperm competition two aspects that have in the literature be examined closely in a wide range of species across different taxa and found to be related to low sperm production (as cited). The intention of these statements was to make a logical suggestion to highlight

something that appears interesting and by highlighting it hopefully encourage someone to do the tests across different species as the reviewer quite rightly points out needs to be done.

The associate editor appears to agree with the reviewer. We have removed the statements, despite that these statements appear to be logical? However, we have left statements on findings in the literature and the findings of the present study. We have also suggested that this would appear to be an interesting area for further research.

It is important to speculate and clearly describe / suggest possible relationships that should be tested and that may stimulate further interesting research. It is also important not to make unsupported conclusions. At times it is fine balance between the two and misunderstandings can follow. The changes made should ensure no misunderstanding, as essentially the logical suggestion that sperm egg-ratio may be linked to reproductive strategies, as has been shown with low sperm production and low sperm competition has been removed.